# Molecular and Morphological Characteristics of the De-Obstructed Rat Urinary Bladder—An Update

**DOI:** 10.3390/ijms231911330

**Published:** 2022-09-26

**Authors:** Bengt Uvelius, Karl-Erik Andersson

**Affiliations:** 1Department of Urology, Skåne University Hospital, Jan Waldenströms Gata 7, SE-21421 Malmö, Sweden; 2Department of Clinical Sciences, Lund University, SE-221 84 Lund, Sweden; 3Department of Laboratory Medicine, Lund University, SE-223 63 Lund, Sweden; 4Wake Forest Institute for Regenerative Medicine, Winston-Salem, NC 27157, USA

**Keywords:** urinary bladder, rat, outlet obstruction, de-obstruction, array, electron microscopy, ultrastructure, units

## Abstract

Many patients with outlet obstruction secondary to prostatic enlargement have lower urinary tract symptoms (LUTSs) and an increased frequency of micturition. The standard treatment is transurethral resection of the prostate (TURP), which alleviates obstruction and symptoms. However, after TURP, 20–40 percent of patients continue to experience LUTSs. The aim of the present study in rats was to identify the mechanisms that do not normalize after the removal of the obstruction and that could explain the persisting symptoms. We had microarray data from control, obstructed, and de-obstructed female rat bladders, which made it possible to study 14,553 mRNA expressions. We also had a bank of electron micrographs from similar detrusors. Microarrays: There were significant differences between the control and obstructed bladders for 1111 mRNAs. The obstructed and de-obstructed bladders differed significantly for 1059 mRNAs. The controls and the de-obstructed bladders differed significantly for 798 mRNAs. We observed many mRNAs that were increased in the obstructed bladder and then decreased to control levels after de-obstruction, and many mRNAs that were decreased in the obstructed bladder and then increased following de-obstruction. mRNAs that were significantly higher or lower in the de-obstructed bladder than in the control bladder were also found. Ultrastructure: The detrusor cells in the obstructed bladders had cross-sectional areas that were much larger than those in the controls. The control cells had smooth outlines and similar cross-sectional areas. The de-obstructed detrusor cells had larger cross-sectional areas than the controls, as well as corrugated surfaces. The cell areas varied, suggesting that the shrinkage of the de-obstructed cells was not even. We did not find any points of contact of the gap junction plaque type between the detrusor cells. There were abundant finger-like processes between the detrusor cells in the obstructed and in de-obstructed bladders, which were only occasionally found in the control detrusors. They are the only possible localization for gap junction channels. The de-obstructed rat bladder is not an organ with properties intermediate between those of the control and obstructed bladders. Instead, de-obstructed bladders have gene expressions, morphologies, and functional properties of the individual cells and their organization, which make them distinctly different from both control and obstructed bladders.

## 1. Introduction

Many patients with outlet obstruction secondary to prostatic enlargement have lower urinary tract symptoms (LUTSs), such as hesitancy during micturition, irritative sensations, and bladder overactivity during the periods between micturitions. The frequency of micturitions may increase. The standard treatment is transurethral resection of the prostate (TURP), which alleviates the obstruction. Although after TURP most patients are relieved from symptoms, 20–40% of them continue to experience LUTSs [1,2,3,4,5].

Functional studies of obstructed bladders are abundant. However, mainly because of ethical restrictions, few studies have focused on *human* bladders [6]. Many animal models of outlet obstruction have been developed [7,8], most of them on female rats [9,10].

Partial outlet (urethral) obstruction of the bladder leads to the enlargement [11] of the detrusor muscle due to both hypertrophy and hyperplasia of the detrusor smooth muscle cells [12,13]. The muscle cells start to synthesize fibrillary proteins, both intra- (e.g., desmin) [14,15] and extra-cellular (e.g., collagen) [16], leading to an increased stiffness of the bladder wall [17]. The spatial density of innervation decreases [18], partly due to the enlargement of the muscle component; the presence in the detrusor of areas devoid of nerves (patchy denervation) has also been observed [19]. Detrusor overactivity (DO) is commonly observed by cystometry in rats with outlet obstruction [10].

The removal of the obstructing ligature rapidly induces a weight loss of the detrusor [20,21] and, in most rats, a disappearance of DO, as revealed by cystometry [22]. Approximately 30 percent of the bladders will still have DO [23,24]. The number of studies on what happens to the bladder wall following de-obstruction is limited. This is unfortunate, as a de-obstructed bladder model might give information on what changes may occur in the human bladder after TURP, and why a considerable proportion of the patients who undergo TURP still have LUTSs.

We had access to microarray data from control, obstructed, and de-obstructed bladders of female rats. This made it possible for us to examine, in a systematic manner, the effects of obstruction and de-obstruction on the mRNA expressions of around 15,000 proteins. These arrays have previously been used [25] to characterize what happens at the molecular level during the growth of an obstructed bladder. We also had at hand a bank of electron micrographs from similar detrusors.

The aim of the present study was to find parameters in the rat bladder that do not normalize after the obstruction is removed and to delineate the mechanisms that could contribute to the persistent overactivity found in some bladders after de-obstruction. These potential mechanisms suggest that, after de-obstruction, the bladder does not return to the “control” condition but instead has properties different from both control and obstructed bladders.

## 2. Results

### 2.1. mRNA Expression in Control, Obstructed, and De-Obstructed Bladders

The arrays show the expression of 14,553 mRNAs. The 6-week obstructed bladders had significantly different expressions of 1111 mRNAs (7.6%) compared with the control bladders. A total of 421 (2.9%) were higher, and 690 (4.7%) were lower in the obstructed bladders.

The 10-day de-obstructed bladders had significant differences compared with the control bladders for 798 mRNAs, corresponding to 5.5%. A total of 618 (4.2%) were significantly increased, and 180 (1.2%) were highly significantly decreased.

Compared with the obstructed bladders, the de-obstructed bladders had (Figure 1) a significant difference for 1059 mRNAs (7.3%). The de-obstructed value was significantly higher for 857 (5.9%) mRNAs, and for 202 (1.4%), it was significantly lower than in the obstructed bladders.

The greatest difference (Table 1) was found for gremlin 1, which has no recognized function in the urinary bladder. The adenosine A2B receptor is upregulated in de-obstructed bladders, while its mRNA level is similar in control and obstructed bladders.

Endothelin mRNA increased two-fold in the obstructed and de-obstructed bladders. The endothelin A receptor also increased significantly in both the obstructed (2.05-fold) and de-obstructed (1.57-fold) bladders.

The 4.4-fold increase in the Cthrc1 mRNA in the obstructed bladders and its 2.2-fold increase in the de-obstructed bladders were statistically significant. The collagen type 1 alpha 1 mRNA was unchanged in the obstructed bladders (0.82) compared with in the controls, but it decreased significantly to 0.493 in the de-obstructed bladders.

There was no overlap between the top 26 list of changes in mRNA for the obstructed and de-obstructed bladders.

### 2.2. Marked mRNA Differences between Obstructed, De-Obstructed, and Control Bladders

#### 2.2.1. mRNAs for Structural Proteins

The mRNA for the intermediate filament protein desmin was significantly increased (1.25) in the obstructed bladders but normalized (1.02) in the de-obstructed bladders.

The mRNAs for collagen type 1 alpha 1 and alpha 2 (fibrillar collagen) were unchanged in the obstructed bladders (0.82 and 0.86, respectively), but they were significantly decreased (0.493 and 0.535, respectively) in the de-obstructed bladders. Collagen type 4 alpha 1 mRNA, residing in the basal lamina in the smooth muscle cells, was significantly increased in the obstructed bladders but normalized in the de-obstructed bladders.

The mRNA for elastin was significantly higher (2.28) in the obstructed bladders. In the de-obstructed group, the mRNA level (0.88) returned to the control level.

#### 2.2.2. mRNAs for Proteins Involved in Metabolism

The glucose transporter Glut1 increased (1.62) significantly in the obstructed bladders, but it normalized (1.12) in the de-obstructed bladders. There was no difference in the Glut 3 mRNA levels in the control, obstructed, and de-obstructed bladders (1.0, 1.35, and 1.12, respectively). The LDH subunit A mRNA level was significantly higher (1.60) in the obstructed bladders, but it normalized (1.10) in the de-obstructed bladders. Ahrr (a dioxin receptor repressor, which inhibits HIF-dependent transcription) significantly decreased in both the obstructed (0.492) and de-obstructed (0.434) bladders.

#### 2.2.3. Neurotrophins

The only neurotrophin mRNA in the arrays that was affected by obstruction was the Trk B receptor agonist brain-derived neurotrophic factor (BDNF). It almost doubled (1.92) in the obstructed bladders. The increase was significant but normalized (1.11) in the de-obstructed bladders. The mRNA for NT-3 and NGF was not affected by obstruction or de-obstruction.

#### 2.2.4. Connexins

The mRNA for connexin 43 increased to 1.79 in the obstructed bladders, but it normalized (0.91) following de-obstruction. In the arrays of the denervated bladders (see Materials and Methods), connexin 43 mRNA increased 3.30-fold in comparison with the controls. The mRNA levels of connexin 26 were the same in the controls, obstructed, and de-obstructed bladders.

#### 2.2.5. Cell Surface Receptors

The endothelin 1 mRNA was significantly increased in the obstructed bladders and remained so after de-obstruction (see above). The endothelin receptor type A mRNA increased significantly (2.05) in the obstructed bladders. The mRNA level remained significantly increased (1.57) during de-obstruction. Bladder denervation induced an even higher level (2.51) of the endothelin A receptor mRNA.

The β3-adrenoceptor mRNA was significantly decreased (0.56) in the obstructed bladders. The de-obstructed bladders were not significantly lower in mRNA compared with the control bladders (0.76).

Among the purinergic receptors, the P2x1 receptor mRNA level was unaffected by obstruction or de-obstruction. However, the P2x5 receptor mRNA was significantly increased in both the obstructed and de-obstructed bladders. P2y1, P2y2, P2y4, and P2y6 mRNAs were not affected by obstruction or de-obstruction. The adenosine A2B receptor mRNA was similar in the control and obstructed bladders, but it was significantly higher (2.11) in the de-obstructed bladders.

The arginine vasopressin mRNA level was significantly decreased (0.53) in the obstructed bladders but normalized in the de-obstructed bladders.

The insulin-like growth factor mRNA was similar in the control, obstructed, and de-obstructed bladders. The same was true for the IGF 1 receptor. The IGF1-binding protein 3, however, decreased to 0.24 for the obstructed bladders and to 0.32 for the de-obstructed bladders.

The α_1D_ receptor mRNA increased significantly (1.67) in the obstructed bladders, but it normalized (1.17) in the de-obstructed bladders.

#### 2.2.6. Apoptosis

Caspase 3, a marker of apoptosis, had similar mRNA levels in the control (1.0) and obstructed (1.12) bladders. In the de-obstructed bladders, the mRNA level increased to 1.37, but the increase did not reach statistical significance.

#### 2.2.7. Inflammation

TGFb2 and TGFb2 mRNAs increased significantly (2.20, and 1.88, respectively) in the obstructed group but normalized (1.21 and 1.032, respectively) in the de-obstructed bladders. COX-2 (PTGS2) mRNA increased significantly (2.37) in the obstructed group but also normalized (1.17) in the de-obstructed group.

### 2.3. Structural Differences between Obstructed, De-Obstructed, and Control Detrusors

Light microscopy showing that the amount of collagen increases in obstructed bladders and normalizes in de-obstructed bladders has already been demonstrated by Jin et al. [24].

Electron microscopy shows that, in obstruction, muscle cell transverse profiles are larger (and this accounts for the substantial increase in cell volume), corrugated, and more irregular than in controls (Figure 2). Many of them present laminar invaginations of the cell membrane, covered by basal lamina material and bearing extensive attachment from muscle filaments. The extracellular material between the muscle cells is more abundant. 

In general, the muscle cell contours of a de-obstructed detrusor have a higher cross-sectional area than those of the controls, as well as a considerably corrugated surface, whereas the control cells have smooth outlines (Figure 3). Often, there are bundles in the de-obstructed wall with cell contours that are much smaller than those in the surrounding muscle, suggesting that the shrinkage of the cells is not even. Large processes emerging from the muscle cells are laminar or finger-like, and they abut on adjacent muscle cells, with a close apposition of the membranes. Although they are occasionally found in control detrusors, they are much more common in de-obstructed and obstructed bladders. To quantitate the difference, we counted the number of finger-like processes on 17 micrographs (magnification 20 K) of control detrusors and 17 micrographs of de-obstructed detrusors. The mean number per micrograph was 0.6 +/− 0.2 (SE) for the controls and 3.4 +/− 0.3 (SE) for the de-obstructed detrusors. The difference was highly significant (*p* < 0.001, Student’s t-test, unpaired data).

The distance between the muscle cells is larger than in the controls and (as in the obstructed detrusor) has abundant extracellular material. The ultrastructural appearance of neuro-muscular junctions is the same in all three groups. No degenerating nerve terminals or muscle cells were noted. Cell-to-cell contacts of the gap junction type were not found.

## 3. Discussion

### 3.1. Structural Changes

Long-term partial urethral obstruction in the rat [9,10] is the most used model of outlet obstruction due to prostatic enlargement in males. The mouse model has gained popularity due to the availability of various transgenic mice. However, unlike in BOO, in males, there seems to be no regulation of TNF-α responsive genes in the obstructed mouse bladder [26]. In our study, we found a significant increase in the mRNA for PTGS2 (COX-2) in the obstructed rat bladders. PTGS2 has TNF-α as an upstream regulator. This suggests that the bladder of rats may be better than that of mice for modeling bladder outlet obstruction in patients.

The stretching of smooth muscle cells in general initiates growth processes, including protein synthesis [27] and the synthesis of DNA [28]. In the bladder, obstruction leads to wall stretching, initially through a mechanism corresponding to the Frank–Starling mechanism in the heart [9]. Long-term obstruction also leads to a re-modelling of the growing bladder. In the rat, obstruction leads to an up to ten-fold increase in bladder or detrusor weight. The weight increase of the detrusor is probably a combination of hypertrophy and hyperplasia of the detrusor smooth muscle cells [12,13]. Both mechanisms can cause residual urine. The residual urine can increase further if the bladder becomes decompensated, e.g., secondary to hypoxia/ischemia of the bladder wall [29]. The hypertrophy of the obstructed rat detrusor muscle cells is considerable. In one study [20], the detrusor muscle cells in an obstructed bladder had a volume of about 8900 μm^3^, a threefold increase compared with about 2800 μm^3^ of control muscle cells. The increased weight of the obstructed detrusor is, to some extent, due to the synthesis of extracellular fibrillary proteins. In a control detrusor, the total amount of collagen is 7 mg [30]. Long-term obstruction increases the collagen content to 26 mg, a more than three-fold increase. Following de-obstruction, collagen content decreases to 18 mg, but not further. None of the inflammation markers we studied was elevated in the denervated bladder, so it is unlikely that an increased collagen content in a de-obstructed bladder indicates an early stage of fibrosis.

The most conspicuous difference between the contours of de-obstructed muscle cells and controls is that the former are corrugated [20,21]. The corrugated surface changes the perimeter/area ratio of the cells [21]. Control cells have a cross-sectional area in their nuclear region of 13 μm^2^ and a perimeter/area ratio of 1.27 μm^−1^. The corresponding numbers for the de-obstructed bladders are 20.5 μm^2^ and 1.26 μm^−1^. Thus, as a consequence of their corrugated surfaces, the de-obstructed detrusor cells have, despite a 50% increase in their cross-sectional area, a similar perimeter/area (or surface/volume] ratio to the controls. In the same study, the detrusor cells of the obstructed bladders had a mean cross-sectional area of 66 μm^2^ and a lower (0.86 μm^−1^) perimeter/area ratio [21]. In this study, the cross-sectional areas of the individual control detrusor cells do not differ substantially (Figure 2). There is a marked difference in the size of the cell contours in the de-obstructed detrusors (Figure 3). Often, there are bundles of cells with small cross-sectional areas close to areas with cells that have large contours (Figure 3). It thus seems that the rate of the shrinkage of the cells is different from one cell to the other. We do not know if the new cells that develop during the obstruction are the first to disappear. We did not find any degenerating muscle cells or nerve terminals, and our marker for apoptosis, caspase 3, had similar mRNA levels in the control, obstructed, and de-obstructed bladders. It has, however, previously been shown that the mRNA for Bnip3, a marker for autophagy, almost doubles during both obstruction and de-obstruction [31].

### 3.2. Functional Changes

A characteristic of an obstructed bladder is spontaneous contractile activity observed in vitro in bladder strips, recorded as small increases in pressure in perfused bladders, and confirmed by in vivo cystometries. The origin of this activity has been extensively discussed and several, not mutually exclusive, theories have been advanced [32]. In vivo, this activity occurs during the storage phase (no activity in the parasympathetic motor nerves). Drake et al. [33] suggested that the contractions were a consequence of the detrusor being organized into component modules, each capable of contracting autonomously. It is reasonable to expect that action potentials are triggered in the detrusor smooth muscle cells during ischemia or in areas of patchy denervation. The action potentials would then spread to other cells in its unit. The bigger the unit, the greater the effect on wall tension and, according to the law of Laplace, on bladder pressure. In this context, it is interesting to note that denervated bladders have an even higher expression of connexin 43 mRNA (see Results) than obstructed bladders. Detrusor muscles from such bladders have in vitro tetrodotoxin-resistant spontaneous phasic contractions with amplitudes up to about 40% of maximal activation [34], suggesting very large units. As there are no nerves, the cooperative action of the detrusor cells most likely occurs through connexin 43 low-resistance channels.

The spontaneous contractions observed in patients during cystometry (involuntary contractions—detrusor overactivity: DO) are often related to urgency and/or incontinence and sometimes to pelvic pain. A link between spontaneous contractions observed in vitro and DO in patients has not been established. Spontaneous contractile activity also occurs [22] in de-obstructed bladder tissue (often exaggerated), and some de-obstructed patients have DO. An interesting question is whether the molecular and morphological mechanisms ultimately leading to DO are the same.

De-obstruction leads to a rapid recovery of normal bladder emptying [22]. After a week, there is no residual urine. There is a short-lasting supersensitivity to carbachol, perhaps related to a decrease in the number of nerve terminals as the detrusor weight decreases. The detrusor muscle cells shrink to a final size of 3700 μm^3^ [20]. The diameter of the urethral lumen normalizes with time [20], and the overactivity that develops during the obstruction disappears in most bladders. However, in 20–30%, overactivity can still be demonstrated after de-obstruction [23,24]. It has been suggested that there is a lasting functional obstruction in a group of de-obstructed bladders and that this could be a factor in post-de-obstruction overactivity [23].

The obstructed detrusor cells can, to a considerable extent, compensate for hypoxia by shifting from oxidative to glycolytic metabolism [31]. The present finding of an increased mRNA level for LDH subunit A fits with this. The shift from LD1 to LD5, which consists of four LDH A units, explains how the obstructed detrusor can retain its contractility in anoxia [35,36]. What is detrimental for the detrusor cells is thus not a lack of oxygen but a lack of glucose. In vitro, a lack of glucose rapidly leads to cell death [37]. The control and de-obstructed detrusors have similar levels of mRNA for LDH subunit A. This indicates that the de-obstructed bladders shifted back to oxidative metabolism.

### 3.3. Structural and Functional Considerations, and Size of Dynamic Motor Units

In the female rat, there are about 6000 motor axons to the bladder from the pelvic ganglia [38]. Every detrusor muscle cell is innervated by at least one nerve ending (varicosity) [39]. The number of detrusor muscle cells in a bladder can be calculated in two ways. One way (the “morphometry” method) is to divide the detrusor weight by the volume of one detrusor cell and to compensate for the 20% extracellular space in the detrusor. Another way (the “DNA” method) is to divide the total amount [20] of DNA in the detrusor by the amount [40] in one cell. Almost 100 percent of detrusor cells are muscle cells [11]. The numbers of muscle cells per motor nerve in the control detrusors are 2880 using the morphometry method (with a detrusor weight of 59 mg) [20] and 2830 using the DNA method. Six weeks of obstruction increases these numbers to 10,200 (morphometry: detrusor weight of 679 mg) and 24,500 (the DNA method). The higher number found with the DNA method could be an overestimation caused by an increased amount of DNA in a population of detrusor cells in the obstructed bladders. Six weeks after de-obstruction, the calculated numbers of muscle cells per motor nerve are 7400 (morphometry; detrusor weight of 204 mg) and 5333 (DNA), 2.6 and 1.9 times above the control levels, respectively.

The approximate sizes of the contractile units can also be shown by their weights. If the detrusor weights above are divided by 6000, a control unit would weigh 0.010 mg. The obstructed detrusor would have units weighing 0.113 mg, and the units of a de-obstructed detrusor would weigh 0.034 mg. Figure 4 shows a tentative organization of motor nerves and muscle cells in the rat detrusor. One motor nerve and its muscle cells would constitute a unit. A muscle cell can, however, be innervated by more than one nerve [38] and can thus belong to several units. Moreover, muscle cells can be coupled electrically to each other by gap junctions [41]. The number of gap junction plaques is low (or non-existent) in rat bladders [11], present study. However, the presence [42] of connexin 43 in rat detrusors indicates gap junction channels between the cells. The obstructed and de-obstructed cells have considerably more finger-like processes than the controls. It is reasonable to expect that these structures harbor the gap junction channels. We think that this unitary arrangement is dynamic. Muscle cells can, due to the dense innervation and cell-to-cell coupling, belong to different units at different points in time. Connexin 26 is reported to be expressed [43] in the urothelium, as well as over-expressed following outlet obstruction. In our study, the bladder mRNA level was not affected by obstruction or de-obstruction.

There were significant differences between the controls and de-obstructed bladders for 798 mRNA expressions and for 1059 mRNA expressions between the obstructed and de-obstructed bladders. These high numbers make the evaluation of the functional consequences difficult, and not all differences between the groups are traceable in the arrays. However, since the obstructed and de-obstructed bladders differ in their gene expressions, it is possible that their responses to drugs aimed at treating bladder disorders, such as OAB and DO, can differ. One example is the limited effect of β_3_-adrenoceptor agonists in patients with outlet obstruction [44,45]. Our obstructed rat model has a decreased mRNA level for β_3_-adrenocceptors, and in vitro, the relaxing effect of β_3_-adrenoceptor agonists on pre-contracted rat detrusor strips is almost non-existent [46]. As the mRNA for β_3_-adrenoceptors is normalized in the de-obstructed bladders, one would expect a more favorable effect on DO by β_3_-adrenoceptor agonists in such bladders. Other examples are the increased α_1D_ receptor mRNA in the bladders and the normalization in the de-obstructed bladders. Others have found increased receptor protein in both groups [47]. This could perhaps be explained by the increase in cell membrane area per unit detrusor weight in the de-obstructed bladders caused by the shrinkage of the cells and by their corrugated surfaces.

## 4. Materials and Methods

### 4.1. Surgery

Female rats were anesthetized with ketamine and xylazine, and the bladder neck and proximal urethra were visualized. A prolene 4–0 ligature was then tied around the proximal urethra and a steel rod (with a diameter of 1 mm). The rod was removed, resulting in a standardized obstruction. The animals were then kept for various periods of time and re-operated on, and the ligature was removed (de-obstruction). After the desired de-obstruction period, the animals were killed, and their bladders were dissected out and used for electron microscopy or microarray experiments.

### 4.2. Electron Microscopy

Female rat urinary bladders (control; 6-week obstructed; and 6 weeks of obstruction followed by 6 weeks of de-obstruction) were dissected out and filled with 1 mL saline per 100 mg bladder weight. They were then fixated in 5% glutaraldehyde in 100 mM Na cacodylate buffer. Segments were dissected, osmicated, dehydrated, block-stained with uranyl acetate, and embedded in Araldite. Sections were cut and examined under an electron microscope.

### 4.3. Microarrays

The microarrays used in this study have been published and are publicly available (GEO accession numbers GSE47080 and 104540). The first array [25] consists of four groups of female rat bladders: (1) control bladders; (2) bladders that had been subjected to a partial outlet obstruction for 10 days; (3) bladders that had been subjected to a partial outlet obstruction for 6 weeks; and (4) bladders that had first been subjected to a 6-week period of partial outlet obstruction and then had the obstruction removed, followed by recovery for 10 days. In the present study, we used the results for the control, 6-week obstructed, and bladders with 6 weeks of obstruction followed by 10 days of de-obstruction. A limited amount of data was also collected from an array experiment, where the control bladders were totally denervated by cryo-ablation of the pelvic ganglia and then emptied manually 2 times per day for 10 days before the bladders were collected [48].

### 4.4. Statistics

The results are presented as mean value ± SEM. To identify the significantly differentially expressed genes between the controls and obstructed and de-obstructed groups in the microarrays, TMEV v. 4.0 software was used and produced q values. A q = 0.01 value was considered significant.

## 5. Conclusions

Our study shows that microarrays can be used as the first step to identify the mechanisms involved in the regression of growth in de-obstructed bladders. They did not reveal which mechanisms are the cause of overactivity in some of the bladders or whether they are the same as those in the obstructed bladders; however, they provided some directions for future systematic studies. Our results show that the de-obstructed rat bladder is not an organ with properties intermediate between those in control and obstructed bladders. Instead, de-obstructed bladders have gene expressions, morphologies, and functional properties of the individual cells, and their organization makes them distinctly different from both control and obstructed bladders.

## Figures and Tables

**Figure 1 ijms-23-11330-f001:**
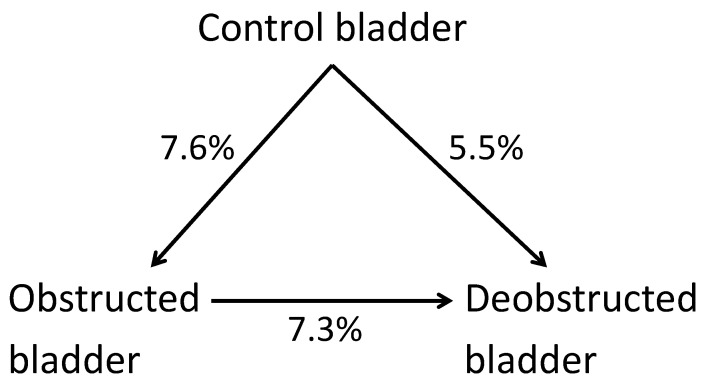
Percentage of mRNAs that differed significantly between arrays of control, obstructed, and de-obstructed bladders, each containing 14,553 mRNAs. It seems that de-obstructed bladders differ as much from control bladders as they do from obstructed ones.

**Figure 2 ijms-23-11330-f002:**
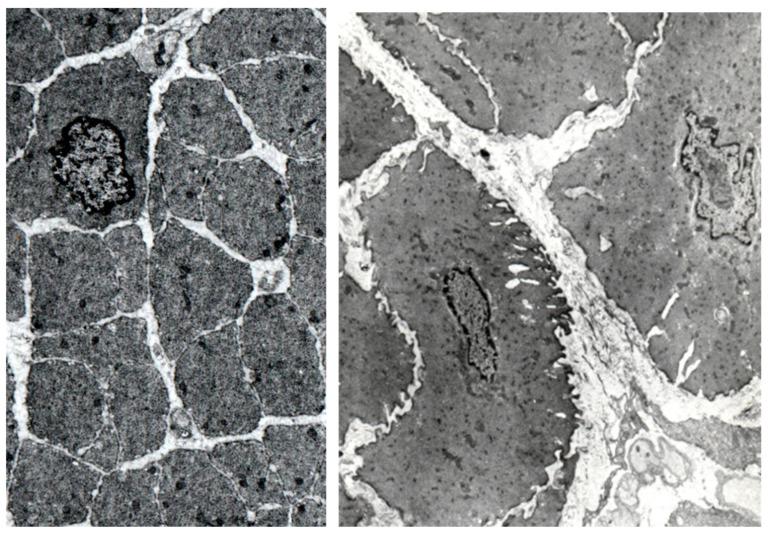
Electron micrographs of a detrusor muscle from a control bladder (**left**) and a 6-week obstructed bladder (**right**). The contours of the smooth muscle cells in the control bladder are circular to ovoid, and the perimeter has a regular outline. The detrusor muscle cells from the obstructed bladder are much larger and have a surface that is full of invaginations. The distance between the cells is often increased. The intercellular space contains collagen fibrils. The bases of both micrographs correspond to 18 μm.

**Figure 3 ijms-23-11330-f003:**
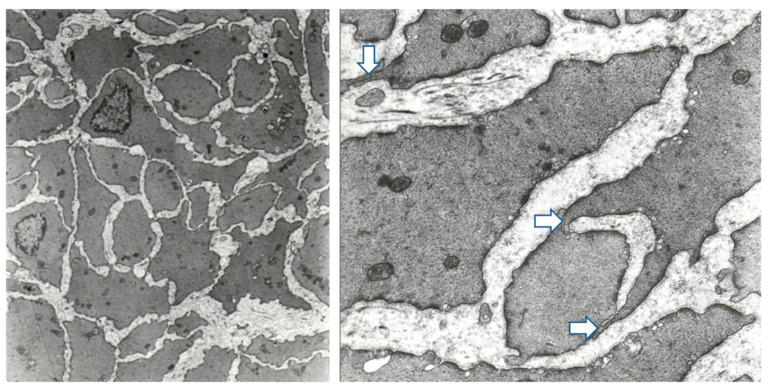
The **left** panel shows detrusor muscle cells from a bladder obstructed for six weeks and then de-obstructed for another six weeks. The outlines of the cells are much more irregular than those of the control bladder (see Figure 2). There is also a more pronounced variation in the cross-sectional area between the different cells. The **right** panel shows finger-like processes (arrows) reaching from one cell to another. There are no other points of contact between the cells. The bases of the **left** and **right** micrographs correspond to 21 and 5.6 μm, respectively.

**Figure 4 ijms-23-11330-f004:**
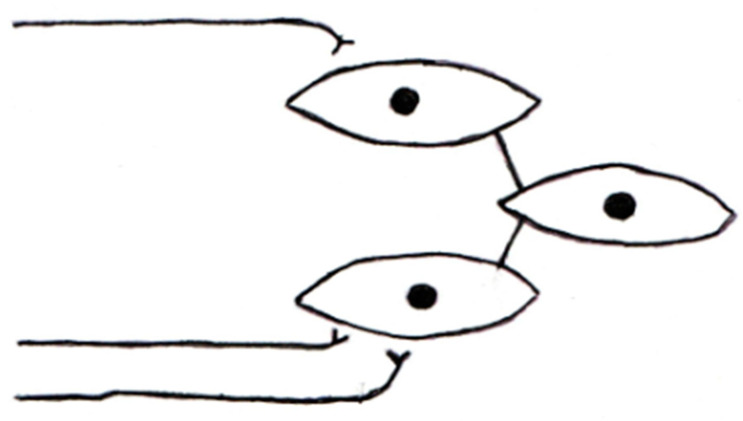
A tentative illustration of the organization of motor nerve fibers from the pelvic ganglia and the detrusor smooth muscle cells. According to the literature, each muscle cell is innervated by one or more neuromuscular junctions. This suggests that each muscle cell can belong to several motor units. Moreover, connexin-43-dependent low-resistance contact points between smooth muscle cells would lead to a situation where muscle cells that are reached by a nerve ending can still be part of a unit. The organization of the units is not constant, as it is for motor units in skeletal muscle, and we therefore suggest to call them “dynamic units”.

**Table 1 ijms-23-11330-t001:** Top 26 fold changes for mRNAs between control and de-obstructed bladders.

Gene Symbol	Gene Description	Fold Change
Grem1	gremlin 1, cysteine knot superfamily, homolog (Xenopus laevis)	17,13979
Trim59	tripartite motif-containing 59	2,722014
Cdkn1c	cyclin-dependent kinase inhibitor 1C	2,672856
Rspo3	R-spondin 3 homolog (Xenopus laevis)	2,502077
RGD1564327	similar to integrin alpha 8	2,322538
Cthrc1	collagen triple helix repeat containing 1	2,212791
Eda2r	ectodysplasin A2 receptor	2,182682
Fxyd6	FXYD domain-containing ion transport regulator 6	2,136691
Adora2b	adenosine A2B receptor	2,105719
Edn1	endothelin 1	2,099241
Acadsb	acyl-Coenzyme A dehydrogenase, short/branched chain	2,090642
Angptl1	angiopoietin-like 1	2,064316
Abcb1b	ATP-binding cassette, sub-family B (MDR/TAP), member 1B	2,057768
Phlda1	pleckstrin homology-like domain, family A, member 1	0,494031
Cadps	Ca++-dependent secretion activator	0,49359
Col1a1	collagen type I alpha 1	0,492934
Nrk	Nik related kinase	0,491153
Mfap2	microfibrillar-associated protein 2	0,469284
Nrg1	neuregulin 1	0,466933
Itm2a	integral membrane protein 2A	0,450881
Slc30a10	solute carrier family 30, member 10	0,445451
Hhip	Hedgehog-interacting protein	0,437063
Ahrr	aryl-hydrocarbon receptor repressor	0,43424
Krt20	keratin 20	0,415004
Fam111a	family with sequence similarity 111, member A	0,403881
Igfbp3	insulin-like growth factor binding protein 3	0,322226

## Data Availability

The microarrays in this study are publicly available (GEO accession numbers: GSE47080 and GSE104540).

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
