# Peer review of "Molecular and Morphological Characteristics of the De-Obstructed Rat Urinary Bladder—An Update"

_ijms, 2022, doi:10.3390/ijms231911330_

Round 1

Reviewer 1 Report

The manuscript titled “Molecular and Morphological Characteristics of the De-Obstructed Rat Urinary Bladder – an Update” by Bengt Uvelius provides us the de-obstructed bladders have gene expressions, morphology, and functional properties of the individual cells and their organization make them distinctly different from both control and obstructed bladders. However, there are some comments for this manuscript to improve the quality of manuscripts.

1)     In the results section, structural differences between de-obstructed and control detrusor were described by electron microscope. Electron microscope was used to observing the subcell microstructure. In general, HE staining was used to observe the tissue structural changes, especially in muscle. Please add this in your results.

2)     As the structural changes are crucial for conclusions of this manuscript, the differences between de-obstructed and control detrusor by electron microscope should have quantitative analysis and statistics.

3)     Please add the ethical approval in your materials and methods section, this is very important for your manuscript.

4)    It is noted that your manuscript needs carefully editing by someone with expertise in technical English editing paying particular attention to English grammer, spelling, and sentence structure so that the goals of this review are clear to read.

Reviewer 2 Report

Excellent work with exploration and identification of changes in pathways or markers linked to LUTS, in particular storage LUTS, which are the conditioning factors of symptoms after prostate surgery.

However, there are some situations that can be improved on this paper. First it has to be clarified, why the use of the female animal model, when LUTS due to obstruction are usually in males. Second, the discussion is somewhat disconnected from the results. We have a discussion of functional alterations, but no functional studies, such as cystomanometry, have been performed on the animals. Furthermore, the main part of the discussion is structural changes at the level of muscle fibers, when the results of the work, although they exist in this respect, are only a relatively simple description of the microscopic aspect. The fundamental point of the results are the changes identified by the expressions of mRNAs. However, only some of these changes are discussed and briefly. The paper would benefit greatly from a broader discussion at this level, because the identified changes are relevant for a clinician. 

The investigation itself is very good. Congratulations.
